# By-Products Valorization: Peptide Fractions from Milk Permeate Exert Antioxidant Activity in Cellular and In Vivo Models

**DOI:** 10.3390/antiox13101221

**Published:** 2024-10-10

**Authors:** Valeria Scalcon, Federico Fiorese, Marica Albanesi, Alessandra Folda, Gianfranco Betti, Marco Bellamio, Emiliano Feller, Claudia Lodovichi, Giorgio Arrigoni, Oriano Marin, Maria Pia Rigobello

**Affiliations:** 1Department of Biomedical Sciences, University of Padova, Via Ugo Bassi 58/b, 35132 Padova, Italy; 2Padova Neuroscience Center (PNC), University of Padova, Via Orus 2, 35129 Padova, Italy; 3Veneto Institute of Molecular Medicine (VIMM), Via Giuseppe Orus, 2, 35129 Padova, Italy; 4Centrale del Latte d’Italia S.p.A., Sede di Firenze, Via dell’Olmatello 20, 50127 Firenze, Italy; 5Centrale del Latte d’Italia S.p.A., Sede di Vicenza, Via Faedo 60, 36100 Vicenza, Italy; 6Institute of Neuroscience, Consiglio Nazionale delle Ricerche (CNR), Viale G. Colombo 3, 35121 Padova, Italy

**Keywords:** circular economy, milk permeate, bioactive peptides, antioxidants, Caco-2 cells, zebrafish

## Abstract

The discarding of agri-food by-products is a stringent problem due to their high environmental impact. Recovery strategies can lead to a reduction of waste and result in new applications. Agri-food waste represents a source of bioactive molecules, which could promote health benefits. The primary goal of this research has been the assessment of the antioxidant activity of milk permeate, a dairy farm by-product, and the isolation and identification of peptide fractions endowed with antioxidant activity. The chromatographic extraction of the peptide fractions was carried out, and the peptides were identified by mass spectrometry. The fractions showed radical scavenging activity in vitro. Moreover, the results in the Caco-2 cell model demonstrated that the peptide fractions were able to protect from oxidative stress by stimulating the Keap1/Nrf2 antioxidant signaling pathway, increasing the transcription of antioxidant enzymes. In addition, the bioactive peptides can affect cellular metabolism, increasing mitochondrial respiration. The action of the peptide fractions was also assessed in vivo on a zebrafish model and resulted in the protection of the whole organism from the adverse effects of acute cold stress, highlighting their strong capability to protect from an oxidative insult. Altogether, the results unveil novel recovery strategies for food by-products as sources of antioxidant bioactive peptides that might be utilized for the development of functional foods.

## 1. Introduction

Food systems and the disposal of agri-food by-products is a pressing problem due to their high environmental impact [1,2]. Therefore, the search for recovery strategies and potential novel use of these by-products can not only lead to a reduction of the actual amount of waste, but also become a source of new products in the perspective of circular economy. In this frame, agri-food waste materials can represent a source of novel bioactive molecules that could be used as ingredients or additives of functional foods with benefits for the consumers and for the food system. Indeed, the so-called functionalization of foods is largely utilized in industry to improve the beneficial properties of the products and to increase their stability and safety [3,4]. The use of animal by-products obtained from milk is a critical problem for dairy companies due to its high environmental impact, and their disposal is strictly regulated by law.

The by-product considered in this research is milk permeate (MP), obtained from the milk ultrafiltration process that is employed for the concentration of milk before its fermentation for subsequent yogurt production. MP can be used for animal feeding (especially pigs), but in the frame of a “zero-waste” economy, its valorization is seen as an urgent need. In particular, the extraction of bioactive components to produce value-added foods could be a useful circular strategy. As MP contains low amounts of fat and proteins, the recovery strategies tend to focus on either the lactose [5] or mineral [6,7] content; however, the peptide fraction could also be recovered from this matrix and may have other applications with respect to the ones exploited so far, also with valorization of its other components, i.e., peptides for its complete use. Thus, our research is aimed at overcoming this gap in the potential complete use and valorization of MP.

In Western countries, because of the progressive aging of the population, the incidence of chronic noncommunicable diseases is increasing [8]. Although multifactorial, these diseases have in common an alteration of the redox balance and an increased inflammatory status [9,10]. Apart from the pharmacological approach, a correct diet can have positive effects on human well-being and even influence disease progression, promoting redox homeostasis and anti-inflammatory effects. Particular attention has been directed to bioactive molecules, such as peptides, carotenoids, phenolic compounds, and galacto-oligosaccharides, able to promote health benefits when present in the diet in a sufficient amount [11,12,13,14]. Regarding bioactive peptides, they are constituted by a maximum of 20 amino acids, and they can be released from the parent protein by treatments inducing protein hydrolysis such as food processing, fermentation, or digestion. The bioactive peptides that have been identified so far hold different properties such as antihypertensive, analgesic, anti-inflammatory, and antioxidant [15]. The specific mechanisms through which these peptides lead to these beneficial effects are an active field of investigation.

In this paper, the focus has been set on the antioxidant effects. It is widely known that redox homeostasis results from a balanced equilibrium between reactive oxygen species (ROS) production and scavenging [16]. Among the different cellular compartments, mitochondria are key organelles for the cellular metabolism, being the site of the Krebs cycle and oxidative phosphorylation. Noteworthily, they are also a major source of ROS. Therefore, these organelles have a crucial role in both the regulation of cellular metabolism and the redox balance. As increased oxidative stress is observed in a plethora of chronic diseases, the introduction of antioxidant molecules in the diet promoting the reduction of reactive species could be beneficial. With respect to the mechanism of action through which these molecules may exert their activity, we have previously observed that various milk-derived bioactive peptides can interact with the Kelch-like ECH-associated protein 1/nuclear factor erythroid 2-related factor 2 (Keap1/Nrf2) pathway, promoting an antioxidant response [17]. Indeed, the Nrf2/Keap1 pathway regulates the antioxidant response by controlling the expression of detoxifying genes, with Keap1 acting as an inhibitor of Nrf2 under normal conditions. Upon oxidative stress or with specific stimuli, Nrf2 is released, translocates to the nucleus, and activates antioxidant gene expression [18]. In addition, the antioxidant activity can be associated with an anti-inflammatory effect acting on the nuclear factor kappa-light-chain-enhancer of activated B cells (NF-κB) [19]. Crosstalk between Nrf2 and the NF-κB pathway occurs through shared regulators and reciprocal inhibition, balancing inflammation and oxidative stress responses [20]. Similar results were obtained with some bioactive peptides derived from zein and kefir [21,22], reinforcing the idea of an intriguing crosstalk between antioxidant and anti-inflammatory properties.

Therefore, the overall goal of the present research is to reduce the environmental impact of the by-product MP, searching for bioactive components endowed with antioxidant and anti-inflammatory properties. To reach this objective, in this paper, a combination of different approaches aimed at identifying the antioxidant components, at dissecting their mechanism of action at the cellular level, and at deciphering their effect in a whole organism have been exploited. Our investigations lead not only to the recovery of the protein fraction from milk permeate but most importantly identify the specific peptide fraction endowed with antioxidant activity observed in both in vitro and in vivo models. Therefore, this fraction can induce beneficial effects that go beyond the nutritional value but may work as a functional antioxidant component of the diet.

## 2. Materials and Methods

### 2.1. Reagents

All chemicals and reagents were purchased from Merck-Fluka-Sigma-Aldrich (Darmstadt, Germany) and had a purity ≥98%.

### 2.2. Evaluation of Antioxidant Activity with the ABTS Scavenging Assay

ABTS^•+^ was generated by reacting 7 mM ABTS (2,2′-azinobis(3-ethylbenzothiazoline 6-sulfonate) (CAT: A1888-1G) with 2.46 mM potassium persulfate and the mixture was maintained at room temperature, in the dark, for 18 h before use [23]. An amount of 0.02 mL of MP or peptide fraction (PF) were treated with 1 mL of 0.08 mM ABTS^•+^. Absorbance decrease was measured spectrophotometrically at 734 nm using a Lambda 2 Spectrometer (PerkinElmer, Waltham, MA, USA). A calibration curve was set up with Trolox C (CAT: 648471, concentration used: 10–120 µM), and the results are expressed as Trolox C equivalent antioxidant capacity (TEAC). The evaluation was performed in triplicate.

### 2.3. Estimation of Antioxidant Activity with the DPPH Scavenging Assay

The evaluation of the antioxidant activity with the DPPH (1,1-diphenyl-2-picrylhydrazyl) scavenging assay was performed using the method described by Citta and co-authors [24]. This assay is based on the reaction of the stable-free radical DPPH with the antioxidant compounds present in the samples, leading to a decrease of absorbance. Briefly, 0.02 mL of MP or PF were diluted in 0.08 mL of water and treated with 0.1 mL of 0.16 mM DPPH (CAT: D9132-1G) dissolved in ethanol. Afterwards, the decrease in absorbance was measured spectrophotometrically at 517 nm using a Lambda 2 Spectrometer (PerkinElmer, Waltham, MA, USA). The evaluation was performed in triplicate. The percentage of antioxidant activity inhibition was calculated as:% DPPH scavenging = (Abs_control_ − Abs_sample_)/(Abs_control_) × 100(1)

### 2.4. Determination of Total Phenolic Content

For the determination of total phenolic content, the method described by Citta and co-authors [24] was employed. Briefly, 1 mL of Folin–Ciocalteau reagent (CAT: F9252) diluted 1:2 with water was added to 1 mL of MP. After 3 min, 2 mL of 10% Na_2_CO_3_ was added, and the samples were incubated for 15 min at room temperature. At the end, the absorbance was recorded at 750 nm using a Lambda 2 Spectrometer (PerkinElmer, Waltham, MA, USA). The evaluation was performed in triplicate. Results are expressed as micrograms of gallic acid equivalents per 100 mL of sample (GAE) after setting up of a calibration curve using gallic acid (CAT: 398225, concentration used: 5–75 mg/mL).

### 2.5. Purification of Peptide Fractions

Peptide-enriched fractions were obtained from MP by solid-phase extraction with a STRATA C18 E cartridge (Phenomenex, Torrance, CA, USA). In the first step, the column was activated with 50 mL of 100% acetonitrile (ACN) and washed with 125 mL of 0.1% trifluoroacetic acid (TFA) aqueous solution. MP (30 mL) was loaded onto the column. PFs were obtained by a discontinuous gradient step of ACN by elution with 5%, 30%, and 50% ACN solutions (50 mL for each elution step). Following this, 5–30% and 30–50% ACN fractions were collected, lyophilized (Freeze Dryer, Edwards, Burgess Hill, UK), and stored at −20 °C until further analysis [25,26]. The 5–30% and 30–50% ACN peptide-enriched fractions were analyzed via reversed-phase high-performance liquid chromatography (RP-HPLC). The lyophilized fractions were dissolved in 2 mL of 0.1% TFA aqueous solution and loaded onto a SNAP KP-C18-HS 12 g column (particle size 50 µm, surface area 400 m^2^/g, pore volume 0.95 mL/g, 90 Å pore diameter; Biotage^®^, Uppsala, Sweden). After an isocratic step at 0% ACN for 10 min, the gradient was linearly increased from 0% to 40% in 24 min and, finally, to 100% in 5 min. Flow rate was set to 12 mL/min, and absorbance was measured by UV detection at 220 nm.

### 2.6. Evaluation of Protein Content

To determine the protein content, PFs were treated with Bradford reagent (0.003% Coomassie Brilliant Blue G-250 (CAT: 1.15444), 10% ethanol, 5.5% ortho-phosphoric acid). After 5 min at room temperature, samples were analyzed spectrophotometrically at 595 nm [27] using a Lambda 2 Spectrometer (PerkinElmer, Waltham, MA, USA). The evaluation was performed in triplicate.

### 2.7. Liquid Chromatography–Tandem Mass Spectrometry (LC-MS/MS) Analysis

The lyophilized fractions were dissolved in 0.1% formic acid (FA) water solution. Then, 1 µg of the obtained solution (corresponding to a volume of 2.5 µL) was subjected to LC-MS/MS analysis using an LTQ-Orbitrap XL mass spectrometer (Thermo Fisher Scientific, Waltham, MA, USA) connected online with a nano-HPLC Ultimate 3000 (Dionex–Thermo Fisher Scientific, Waltham, MA, USA). Briefly, the solution was loaded onto a 10 cm pico-frit chromatographic column (75 µm internal diameter, 15 µm tip, New Objective, Littleton, MA, USA) packed in-house with C18 material (Aeris Peptide 3.6 µm XB-C18, Phenomenex, Torrance, CA, USA) at a flow rate of 8 µL/min. Peptides were separated at a flow rate of 250 nL/min with a gradient of ACN/0.1% FA increasing linearly from 3% to 40% in 20 min. Source temperature was set at 200 °C and capillary voltage at 1.2 kV. A data-dependent acquisition (DDA) method was applied: the instrument performed a full scan at high resolution (60,000) in the Orbitrap followed by the MS/MS fragmentation on the ten most intense ions acquired with collision-induced dissociation (CID) in the linear trap.

### 2.8. Database Search and Peptide Identification

Raw data files were analyzed with the Proteome Discoverer software (version 1.4, Thermo Fisher Scientific, Waltham, MA, USA) connected to the Mascot Search engine (version 2.2.4, Matrix Science, London, UK). Protein and peptide identifications were conducted against the bovine section of the UniProt database (version October 2020, 37,517 sequences), using the following parameters: no enzyme, precursor tolerance 10 ppm, and fragment tolerance 0.6 Da. Furthermore, oxidation of methionine was considered as a variable modification. The algorithm Percolator was used to assess the false-discovery rate (FDR), and results were filtered using an FDR < 0.01 both at the protein and peptide.

### 2.9. Cell Culture

The colon cancer cell models Caco-2 were grown in adhesion at 37 °C in a 5% carbon dioxide atmosphere, using high-glucose Dulbecco’s modified Eagle’s medium (DMEM) containing glutaMAX and supplemented with 10% fetal calf serum and 1% Pen-Strep (Thermo Fisher Scientific, Waltham, MA, USA).

### 2.10. Cell Viability

Caco-2 cells (1 × 10^4^) were seeded in 96-well plates and 48 h later treated with 0.05 mg/mL PFs. After 6 h, 180 µM tert-butyl hydroperoxide (TbOOH) was added to induce oxidative stress. After 24 h from peptide addition, the medium was removed and 3-(4,5-dimethylthiazol-2-yl)-2,5-diphenyltetrazolium bromide (MTT CAT: 475989) solution (0.5 mg/mL) in phosphate-buffered saline (PBS) was added for 3 h in the dark at 37 °C. Afterwards, the MTT solution was removed and 100 µL/well of isopropanol/DMSO (9:1) was added to solubilize formazan crystals. The absorbance was measured at 595 and 690 nm using a plate reader (Tecan Infinite^®^ M200 PRO, Männedorf, Switzerland).

### 2.11. Estimation of ROS Production

ROS production in Caco-2 cells was measured by using the probe 5-(and 6)-chloromethyl-20,70-dichlorohydrofluorescein diacetate (CM-H_2_DCFDA CAT: C6827) to monitor cytosolic H_2_O_2_ production. Briefly, cells (5 × 10^3^) were grown in a 96-well plate for 48 h and then treated with 0.05 mg/mL PFs for 24 h. Cells, washed in Hanks’ Balanced Salt Solution (HBSS)/10 mM glucose, were loaded with 10 µM CM-H_2_DCFDA for 20 min in the dark at 37 °C. Subsequently, the fluorescent probe was removed, and cells were rinsed with HBSS/10 mM glucose and subjected to oxidative stress in the presence of 300 µM TbOOH. Fluorescence increase was followed at 485 nm (excitation) and 527 nm (emission) for 90 min using a plate reader (Tecan Infinite^®^ M200 PRO, Männedorf, Switzerland).

### 2.12. Nrf2 and Nf-κB Nuclear Translocation

In order to investigate the Nrf2 and Nf-κB signaling pathways, the translocation of the two transcription factors to the nucleus was followed. For this purpose, nuclear and cytosolic fractions were divided according to the method described by Yao et al., with some modifications [17,28]. Briefly, Caco-2 cells (1 × 10^6^) were grown in T25 flasks for 48 h and then treated with 0.05 mg/mL PF. After 24 h, cells were rinsed with 1 mL of PBS and lysed for 15 min on ice with 100 µL of buffer containing 10 mM Hepes/Tris pH 7.9, 0.1 mM EGTA, 0.1 mM EDTA, 0.1 mM phenylmethylsulfonyl fluoride (PMSF), 10 mM KCl, 1 mM NaF, and a protease inhibitor cocktail (Complete, Roche^®^, Basel, Switzerland). The samples were rapidly added of IGEPAL (5% final concentration), mixed for 15 s and centrifuged at 1000× *g* for 10 min at 4 °C. The pellet (nuclear fraction) was dissolved in 20 mM Hepes/Tris (pH 7.9), 1 mM EGTA, 1 mM EDTA, 0.4 M NaCl in the presence of 0.1 mM PMSF, 1 mM NaF, and protease inhibitors. Samples were mixed every 2 min for 10–15 s and centrifuged at 20,000× *g* for 10 min at 4 °C to discard the debris. Nuclear proteins (30 µg), evaluated according to Lowry et al. [29], were subjected to SDS-PAGE (10%) and subsequently to Western blot (WB) analysis to define the protein expression level. Densitometric analysis of WB was carried out using NineAlliance software (Mini 9 17.01 version, Uvitec Alliance, Cambridge, UK). Proliferating cell nuclear antigen (PCNA) was used as loading reference.

### 2.13. Western Blot Analysis of Antioxidant Enzymes Expression in Treated Caco-2 Cells

Cells were seeded in a 6-well plate (4 × 10^5^ cells/well). Forty-eight hours later, they were treated with 0.05 mg/mL of the PF. The following day, cells were harvested, washed with 1 mL of PBS, and then lysed with 150 μL of ice-cold radioimmunoprecipitation assay (RIPA) lysis buffer containing 150 mM NaCl, 1% Triton X-100, 0.1% SDS, 0.5% DOC, 1 mM NaF, 1 mM EDTA, 5 mM Tris/HCl (pH 7.4), 0.1 mM PMSF, and protease inhibitor cocktail. The amount of protein was determined by the Lowry method [29]. The expression levels of thioredoxin reductase 1 (TrxR1), thioredoxin 1 (Trx1), thioredoxin reductase 2 (TrxR2), thioredoxin 2 (Trx2), peroxiredoxin 1/2 (Prx1/2), peroxiredoxin 3 (Prx3), glutamate–cysteine ligase catalytic subunit (γ-GCSc), glutathione reductase (GR), glutathione peroxidase 1/2 (GPx1/2), superoxide dismutase (SOD1), glyceraldehyde 3-phosphate dehydrogenase (GAPDH), and β-actin were determined by WB. In particular, cell lysates (30 μg of proteins) were subjected to SDS-PAGE (4–12%), then blotted onto a nitrocellulose membrane and probed with the selected primary antibodies. The WB detection was performed using UVITEC (Alliance Q9 Advanced) equipment. Densitometric quantification was performed using NineAlliance software.

### 2.14. Analysis of the Mitochondrial Respiration in Treated Cells

Cellular respiration was determined with the Seahorse XFe24 Analyzer (Agilent Technologies, Santa Clara, CA, USA) following the Cell Mito Stress Test protocol. Caco-2 cells were seeded at a density of 2 × 10^5^ cells/well and grown in complete medium. Afterwards, cells were treated with 0.05 mg/mL of the PF for 24 h. Before the start of the experiment the medium was replaced with XF DMEM Assay Medium (pH 7.4), supplemented with 10 mM glucose, 1 mM sodium pyruvate, and 2 mM glutamine and the cells were subjected to the oxygen consumption analysis at 37 °C. Three measurements were performed of the basal respiration and after the sequential injections of 1 μM oligomycin, 0.5 μM carbonyl cyanide-p-trifluoromethoxyphenylhydrazone (FCCP), and the combination 1 μM antimycin A + 1 µM rotenone with 2 min of mixing in between measurements. For data normalization, after the experiment, cells were lysed with 50 µL of RIPA buffer composed of 150 mM NaCl, 50 mm Tris/HCl, 1 mM EDTA, 1% Triton X-100, 0.1% SDS, 0.5% sodium deoxycholate, 1 mM NaF, and 0.1 mM PMSF and subjected to protein estimation as described by Lowry [29].

### 2.15. Glycolysis Stress Test Assay

Caco-2 cells were seeded at a density of 2 × 10^5^ cells/well in complete medium. Then, cells were treated with 0.05 mg/mL of the PF for 24 h. Afterwards, the medium was changed to XF DMEM Assay Medium (pH 7.4) supplemented with 2 mM glutamine, and cells were incubated in a non-CO_2_ incubator at 37 °C for 1 h before performing the assay. The Seahorse XFe24 Analyzer was calibrated, and the assay was performed using glycolytic stress test assay protocol as suggested by the manufacturer (Agilent Technologies, Santa Clara, CA, USA). Sequential injection of glucose (10 mM final), oligomycin (1 µM final), and 2-Deoxy-D-glucose (2-DG) (50 mM final) were performed, and three measurements were executed after each injection with 2 min of mixing in between measurements. For data normalization, after the experiment, cells were lysed with 50 µL of RIPA buffer composed by 150 mM NaCl, 50 mm Tris/HCl, 1 mM EDTA, 1% Triton X-100, 0.1% SDS, 0.5% sodium deoxycholate, 1 mM NaF, and 0.1 mM PMSF and subjected to protein estimation as described by Lowry [29].

### 2.16. Effects of the PF In Vivo on Zebrafish Larvae

Zebrafish larvae were raised in fish-water at 28 °C until 5–6 d post fertilization (dpf) on a 12 h light/12 h dark cycle using standard procedures. On the day of the experiment, the larvae were divided into two groups: a control group maintained at 28 °C and an experimental group placed in 10 °C water for 5 min. The latter group was further divided into 3 experimental groups: an acute group tested immediately after the cold stress, and two recovery groups tested after a 30 min recovery period. One of the two recovery groups received the 5–30% ACN PF treatment at a final concentration of 0.05 mg/mL in the cold water. For each test, a standard 24-well plate was utilized with one larva placed in each well. Each plate contained the different experimental groups and the control group. The test was conducted using a DanioVision system (Noldus, Wageningen, The Netherlands). The protocol included an initial phase of light lasting 10 min, followed by alternating periods of light and darkness, each lasting 10 min. Python was used to preprocess the DanioVision datasets.

### 2.17. Statistical Analysis

The indicated values are the mean ± SD of at least three independent experiments. The analysis of variance was performed by multiple comparison test with Tukey–Kramer method, using the software GraphPad InStat 3. Only differences with *p* < 0.05 were considered significant.

## 3. Results

The composition of milk permeate (MP) is reported in Table 1 and was performed in outsource (CHELAB, Resana, TV, Italy). Briefly, proteins were evaluated with the Dumas combustion method [30], total fats were determined according to Baldini et al. [31], lactose was tested with HPLC analysis against proper standards. Regarding vitamins, the quantification of vitamin B1 (thiamine) was assessed fluorimetrically and the quantification of vitamin B2, D2, E, and carotenoids were also performed with HPLC analysis but, if present, were lower than the limit of detection (LoD: B2: 0.10 mg/100 g; D2: 1 µg/kg; E: 0.10 mg/100 g; carotenoids: 0.30 mg/kg).

From this first exploratory analysis, MP appears as a diluted matrix principally constituted by lactose. However, a certain amount of proteins, although limited, can also be detected. In addition, following the process of milk ultrafiltration, the lipophilic vitamins are absent and retained in the milk concentrate with the lipid fraction, while among the hydrophilic vitamins, thiamine is still present in the MP.

### 3.1. In Vitro Estimation of the Antioxidant Activity of Milk Permeate

At this point, the free radical scavenging activity of MP was assessed in order to check whether it could be a source of antioxidant molecules. Several in vitro techniques have been used for preliminary testing of the antioxidant capacities of MP. In particular, assays based on the scavenging of the radicals 2,2′-azinobis(3-ethylbenzothiazoline-6-sulfonic acid) (ABTS) and 1,1-diphenyl-2-picrylhydrazyl (DPPH) were performed. In addition, as high phenolic content has been linked to an elevated antioxidant capacity, the Folin–Ciocalteu assay was also carried out.

For the ABTS scavenging assay, a calibration standard curve with Trolox C was set up, and the data are thus expressed as equivalents of Trolox C (TEAC). The DPPH assay instead is reported as percentage of DPPH scavenging (see Section 2 for further details). Both methods highlight a discrete antioxidant activity of MP as a free radical scavenger (Table 2). Of note, the different chemistry occurring in the two assays utilized for estimating the total radical quenching activity cannot be directly compared, as each radical exhibits different reaction rates with MP components.

Based on these promising results highlighting an antioxidant effect and with the aim of identifying the molecules responsible for the observed antioxidant activity, the total phenolic content was investigated. The data are reported in Table 2 as gallic acid equivalent (GAE). From the results, it is apparent that phenolic compounds are present in the MP and may account at least partially for the observed antioxidant activity of the studied matrix.

### 3.2. Chromatographic Purification of Peptides from Milk Permeate and Identification of Their Sequences

The observed antioxidant activity of MP could derive from bioactive peptides that can gather in the MP during ultrafiltration and originating from the proteolytic cleavage of caseins and other milk proteins. In fact, in the last decades, different studies recognized that milk proteolysis can release short peptides endowed with antioxidant activity [17,32,33]. Thus, the peptide fractions (PFs) were isolated from MP taking advantage of a chromatographic separation via solid-phase extraction operating a discontinuous gradient of acetonitrile (ACN) (more details in the Section 2). Upon lyophilization, a good yield of peptides was obtained in the 5–30% ACN and in the 30–50% ACN fractions. To acquire a higher resolution in peptide separation, the two selected fractions were further purified by RP-HPLC using a continuous gradient of ACN. From the chromatographic separation, several peptides with different abundance were observed in both PFs, suggesting an effective peptide enrichment (Appendix A).

Afterwards, the peptides present in the different fractions were identified via LC-MS/MS analysis. Proteome Discoverer and Mascot software were used to identify the sequence of the peptides, which are reported in Appendix A for the 5–30% ACN and in the 30–50% ACN PFs, respectively.

This analysis is fundamental to characterize the peptide composition of MP. Of note, several peptides were identified, and the ones reported have a length from 7 to 14 amino acids. Indeed, it is well known that the beneficial properties of the bioactive peptides depend on their specific amino acid composition and sequence. In addition, most of them are formed by a limited number of amino acids [34]. As one can notice, the majority of them derive from α and β caseins, but it was also possible to identify peptides from other milk proteins such as lactoperoxidase, lipoprotein lipase, glycosylation-dependent cell adhesion molecule 1 (GLCM1), and polymeric immunoglobulin receptor.

### 3.3. Antioxidant Activity of PFs In Vitro

After the identification of the specific composition of the isolated fractions in terms of peptides, the research moved to the investigation of their antioxidant properties in vitro to determine whether the antioxidant activity of MP could be ascribed to the peptides and to determine which fraction was endowed with the strongest antioxidant activity.

Hence, the ABTS and DPPH radicals quenching assays and the total phenolic content determination were carried out on both the 5–30% ACN and on the 30–50% ACN PFs.

The results are depicted in Figure 1 and clearly indicate that the 5–30% ACN PF has a greater antioxidant capacity. In fact, its radical scavenging activity is higher than the 30–50% ACN PF especially in the ABTS test (Figure 1A), but also in the DPPH scavenging assay (Figure 1B). Furthermore, as shown in Figure 1C, the same fraction presents a higher phenolic content in agreement with the greater antioxidant capability.

Altogether, these data point out that the antioxidant properties can be correlated, at least in part, to the different hydrophobicity of the peptides collected in the two fractions, with the 5–30% ACN PF being the most active in all the three assays performed.

### 3.4. Antioxidant Activity of PFs in Caco-2 Cells

Given the positive results obtained with the radical scavenging tests pointing out an antioxidant activity of the two PFs, we then moved to assess whether they retain this activity in the cellular environment using the Caco-2 intestinal cell model. This specific cell line was selected as it is a model of intestinal cells, which are responsible for the absorbance of the bioactive components from the diet.

First, the effect of the two PFs on the cell viability was determined via the MTT test in control conditions and in the presence of an oxidative insult given by the addition of tert-butyl hydroperoxide (TbOOH). As shown in Figure 2A, the treatment with the PFs for 24 h per se does not harm the cultured cells (grey bars). When TbOOH was added (blue bars), it is possible to discriminate a different action of the two fractions. In particular, the treatment with just the pro-oxidant compound led to a spare viability of 20%, which was not rescued by the presence of the 30–50% ACN PF. On the other hand, the 5–30% ACN PF was able to partially protect cells from the oxidative damage in a concentration-dependent mode. This first result confirms the ability of the 5–30% ACN PF to exert an antioxidant effect also in the cellular context. The fact that only the 5–30% ACN PF is effective in protecting from oxidative stress induction could be due to its greater antioxidant activity already seen in the previous experiments.

Besides the effect on cell viability, to determine the antioxidant activity of the PFs in cells, we tested whether they influence the cellular production of reactive oxygen species (ROS) in basal conditions or after oxidative stress induced by TbOOH. To obtain these results, we employed CM-H_2_DCFDA, a cell-permeable fluorogenic probe that is useful for the detection of cytosolic ROS, especially H_2_O_2_. Both PFs did not alter the ROS levels by themselves with respect to the control (Figure 2B, grey bars). Instead, as expected, the incubation with TbOOH led to a rise of ROS species of about 40%. Among the two fractions, only the 5–30% ACN PF was able to counteract the effect of the oxidant, decreasing the ROS production at both concentrations tested, in accordance with what was seen in the cell viability assay.

Therefore, with these first explorations in the cell model, we can assert that the 5–30% ACN PF exerts an antioxidant effect in cells, being able to contrast the decrease in cell viability and the overproduction of ROS species induced by a well-known pro-oxidant agent. Considering these data, we also decided to choose the 5–30% ACN PF, which was the one with the greatest antioxidant activity, for the next experiments exploring its mechanism of action.

### 3.5. Effects of the PF on the Nuclear Translocation of Transcription Factors in Caco-2 Cells

To dissect the mechanism of action of the 5–30% ACN PF, we examined its potential effect on the antioxidant signaling pathway Keap1/Nrf2. Nrf2 is usually confined in the cytoplasm in a complex with Keap1. However, an oxidative insult or electrophiles can induce the dissociation of the complex and the translocation of the transcription factor (TF) Nrf2 to the nucleus, where it binds to the antioxidant response elements, promoting the expression of antioxidant and Phase II enzymes [18]. We previously reported that some peptides isolated from milk were able to induce Nrf2 nuclear translocation and its signaling cascade [17]. Thus, upon cell treatment with the PF, the nuclear translocation of Nrf2 was measured.

To do that, after treatment with the PF and/or with N-acetylcysteine (NAC), a known antioxidant molecule, we extracted the cellular nuclear fraction from Caco-2 cells and quantified the content of Nrf2 as a readout of activation of the Keap1/Nrf2 pathway. The proliferating cell nuclear antigen (PCNA) was employed as a loading control. In addition, as a functional crosstalk between the Nrf2/Keap1 and the NF-κB pathway is known [20], we also evaluated the nuclear translocation of this pro-inflammatory TF.

In Figure 3, it can be noted that the PF is able to induce Nrf2 nuclear translocation, suggesting the activation of an antioxidant response, and, at the same time, decreases NF-κB translocation, indicating an anti-inflammatory effect in the treated cells. These actions are not due to a general effect of the PF as a radical scavenger, as the antioxidant molecule NAC is unable to induce such effects on the two TFs. In addition, after the combined treatment of the cells with the PF and NAC, it is still possible to appreciate an increased Nrf2 nuclear translocation.

### 3.6. Effect of the PF on the Expression of Antioxidant Enzymes in Caco-2 Cells

The promotion of Nrf2 nuclear translocation and its activity on the antioxidant response elements leads to the transcription of many genes coding for antioxidant enzymes involved in cell protection against oxidative stress. Therefore, we evaluated the protein levels of different genes regulated by this factor by Western blot analyses. The specific genes considered belong to the glutathione system as glutamate–cysteine ligase catalytic subunit (γ-GCSc), glutathione reductase (GR), and glutathione peroxidase 1/2 (GPx1/2). In addition, we checked for superoxide dismutase (SOD1) and enzymes of the thioredoxin system such as thioredoxin reductase 1 and 2 (TrxR1 and TrxR2), thioredoxin 1 and 2 (Trx1 and Trx2), and peroxiredoxin 1/2 and 3 (Prx1/2 and Prx3).

Experimentally, cells were treated with the PF for 24 h, then lysed and subjected to Western blot analysis as described in the Section 2. Glyceraldehyde-3-phosphate dehydrogenase (GAPDH) and β-actin were used as loading controls. The results obtained are reported in Figure 4A–C. The different proteins tested showed an increased expression in the PF-treated cells when compared to the basal condition, even if most of them do not reach a statistical significance. Of interest, the expression of γ-GCSc and GPx1/2 from the glutathione system and Trx2 from the thioredoxin system show the greatest overexpression in Caco-2 cells upon treatment with the PF. The significance of this pattern of expression needs to be further analyzed, but it confirms that the exposure of the cell to the PF elicits an antioxidant response, which likely increases the capacity of the cell to cope with oxidative insults.

### 3.7. Impact of the PF on the Cellular Metabolism

Next, the effect of the 5–30% ACN PF on the overall cellular metabolism was assessed particularly in relation to the mitochondrial activity. Indeed, mitochondria are metabolic organelles, but they are also central cellular ROS producers and thus involved in redox signalling. To this aim, cells treated with the PF for 24 h were subjected to the analysis of the oxygen consumption and the extracellular acidification rates to determine the cellular metabolic status in relation to oxidative phosphorylation and glycolysis (Figure 5).

For what concerns the mitochondrial oxygen consumption, in Figure 5A,Aˈ, we can see that the PF induces an increase in the cellular respiration, yet the differences in comparison to the control appear significant only for the basal respiration rates. This result may indicate that this increase in oxygen consumption does not alter the general respiratory capacity of the cell (both mitochondrial and non-mitochondrial), but more likely predispose and arrange the cell to produce more ATP from oxidative phosphorylation. Accordingly, when the glycolytic rates were measured (Figure 5B,Bˈ), no differences were observed upon cell treatment with the PF highlighting a prominent effect on redox signaling pathways rather than on cellular metabolic settings. Hence, these data suggest that the PF stimulates mitochondrial activity resulting in a positive effect on cellular redox homeostasis.

### 3.8. In Vivo Model: Effects of the PF on Zebrafish Larvae

At last, we assessed whether the antioxidant activity shown by the PF in in vitro experiments could be recapitulated in an in vivo model. To this aim, we exploited the cold stress assay in Zebrafish (*Danio rerio*) larvae, since it is known that cold stress can increase ROS production and oxidative stress promoting inflammatory responses in this animal model [35,36,37,38,39]. Basically, larvae were raised in water at 28 °C and, on the day of the experiment, they were divided into two groups: a control group maintained at 28 °C and an experimental group placed in 10 °C water for 5 min (cold stress). The latter was further subdivided in three groups: an acute group tested immediately after the cold stress, one group tested after a 30 min recovery, while the third group was analyzed after a 30 min recovery in presence of the PF in the fish water. The test consists in tracking the activity and the swimming behavior of the larvae in the dark and in lighted environment (see Section 2).

From Figure 6A,B, it is possible to notice that larvae subjected to cold stress exhibited suppressed locomotor activity, which was partially restored after the recovery period. However, it is only in the presence of the PF that the larvae fully recovered from cold stress even exceeding the performances of the control group. This latter result is particularly relevant for the translatability potential of this research as it highlights the capacity of the PF of working against an oxidative insult also in vivo.

## 4. Discussion

Until recently, milk permeate was discarded as waste or employed as animal feed, but in the frame of the circular economy concept, by-products are being recognized for their potential use as sources for the production of value-added products. The use as it is generally is addressed at exploiting lactose; however, the use of the whole matrix can have some disadvantages due to the other components [40].

This research was developed with the main purpose of a potential valorization of MP, for the extraction of bioactive components that could induce health benefits beyond their nutritional effects when introduced in the diet.

Among the different classes of bioactive molecules our attention has been focused on bioactive peptides. In the last decades, the biological activity of bioactive peptides has been thoroughly explored and several effects including antimicrobial, antihypertensive, anti-inflammatory, anticancer, and antioxidant have been reported [15]. This latter effect is a crucial field of investigation as many non-communicable diseases are linked with an increased oxidative stress and thus antioxidant nutritional interventions could be beneficial [41]. Indeed, the intake of antioxidants is inversely correlated to the aging process and to the development of diseases such as neurodegenerative diseases and diabetes [42,43,44,45]. Furthermore, in the food development, the addition of antioxidant peptides could also reduce oxidative changes over time, prolonging the shelf-life of the product [46].

We started evaluating the antioxidant activity of MP in vitro using radical scavenging methods. As reported in Table 2, a certain antioxidant activity was in fact present. In a previous report that measured the TEAC in whole milk, serum, and whey, it was observed that the ABTS scavenging capacity was several-fold higher in milk than in its fractions [47]. This finding was later confirmed by another paper in which the authors stated that caseins, which are almost absent in serum and whey, were quantitatively the major radical scavenger species present in whole milk [48]. Accordingly, our results show a lower antioxidant activity of MP with respect to what is reported in the literature for whole milk as it does not contain molecules with a molecular weight above 10 kDa. With respect to the phenolic content, in accordance with what we observed in the scavenging tests, the values are about one-eighth with respect to the values reported for raw cow milk (420–490 mg GAE/100 mL) [49] indicating that also in this case antioxidant molecules are still present in MP. The analysis of the overall composition of MP reported in Table 1 led us to hypothesize that these phenolic compounds titrated in the assay could derive from the aromatic amino acid residues of peptides present in the matrix. Hence, the peptide composition of MP was analyzed.

Even though the protein content was limited, the chromatographic separation via solid-phase extraction, obtained two purified peptide fractions (5–30% and 30–50% ACN PF) that were later characterized for their peptide composition. The peptides present were identified via LC-MS/MS analysis (Appendix A). Most of them derive from α and β caseins but also other milk proteins such as lactoperoxidase, lipoprotein lipase, GLCM1, and polymeric immunoglobulin receptor originate some peptides characterizing MP.

Several studies have displayed the significant impact of peptides on the antioxidant activity of milk-derived products [50]. Hence, we performed the free radical scavenging tests and the analysis of the phenolic content on the isolated PFs. Both PFs were active as free radical scavengers, but interestingly, the 5–30% ACN PF was far more effective especially in the ABTS scavenging test (Figure 1A). Likewise, its phenolic content was greater than the 30–50% ACN PF (Figure 1C). These results suggest that the bioactive peptides endowed with antioxidant activity were predominantly isolated within the 5–30% ACN PF.

These assays based on in vitro reactions, although useful as first screenings for the assessment of the antioxidant activity, needed validation in a biological setting. Consequently, we decided to move to a cell model employing the intestinal cell line Caco-2.

In the cellular environment, the capacity of the two PFs to protect from oxidative stress was investigated. As displayed in Figure 2, the 5–30% ACN PF was able to contrast both the cytotoxic effect and the promotion of reactive species production induced by TbOOH, while the 30–50% ACN PF was ineffective. These results in cell culture confirmed the antioxidant activity of the 5–30% ACN PF and prompted us to further explore the mechanism of action leading to the observed activity.

In a previous study, we showed that some milk-derived bioactive peptides exerted their antioxidant action through the activation of the Keap1/Nrf2 axis, which regulates redox homeostasis [17,51]. Their mechanism involved the disruption of the association between Keap1 and Nrf2, with the consequent Nrf2 nuclear translocation and promotion of the transcription of antioxidant and phase II genes. The results shown in Figure 3 indicate that there is an increased translocation of the TF to the nucleus following cell incubation with the PF, indicating an effect on this antioxidant pathway. In addition, as Nrf2 and NF-κB pathways are closely related and since we already identified some bioactive peptides acting both as antioxidant and anti-inflammatory [19], we also checked for NF-κB nuclear translocation. As shown in Figure 3, the PF reduces the amount of this pro-inflammatory TF in the nucleus. Consequently, the PF can act on both these pathways leading to an antioxidant and anti-inflammatory effect. This result could be particularly relevant for the future use of these peptides in the diet of patients affected by age-related diseases that often associate increased oxidative stress to a chronic inflammatory status [41].

To keep investigating the mechanism of action of the PF, we then evaluated the expression of antioxidant enzymatic proteins in the treated cells by Western blot analysis. Intriguingly, a specific pattern of expression emerged in which the most overexpressed enzymes were γ-GCSc, which is the rate-limiting enzyme in the biosynthesis of glutathione, and GPx1/2 that uses glutathione for detoxification from hydroperoxides. Another enzyme showing an important overexpression in cells treated with the PF is Trx2. This small protein resides in the mitochondrial compartment and belongs to the other major cellular thiol antioxidant system, namely the thioredoxin system. Its activity as a thiol disulfide reductase is involved in the control of the redox homeostasis in mitochondria, which are a major source of ROS in the cell. This distinct set of overexpression could also be due to the concerted intervention of other TF in response to the PF exposure, which can be explored in subsequent research. Notably, these results confirm that the PF can elicit an antioxidant response increasing the capacity of the cell to deal with oxidative stressors.

Mitochondria are an important source of cellular ROS and are the site of oxidative phosphorylation and aerobic metabolism. Of note, in recent years, particular attention has been focused on the study of how diet influences metabolic pathways. Therein, we also investigated the effects of the 5–30% ACN PF on the cellular oxygen consumption and glycolytic rates via extracellular flux analyses. From the data shown in Figure 5, it is possible to observe that the PF increases the basal oxygen consumption rates while no major effects were elicited on the glycolytic rates. The higher oxygen consumption in comparison with the untreated control cells in the basal condition suggests a rise in ATP production by the oxidative phosphorylation. The identification of the specific mechanism and the relevance of these effects on the cellular metabolism open an interesting new branch of investigation that go beyond the scope of the present paper but already show a targeted action of the PF on mitochondria.

Finally, the effects found in the animal model are of considerable importance as we showed that the PF was also effective in vivo in counteracting the stress effects elicited in zebrafish larvae exposed to cold water, which is known to induce oxidative stress in fish [35,36,37,39,52]. The larvae motility, highly reduced after cold stress, was rescued in the presence of the PF (Figure 6), indicating an effective action of the PF against the stress. In addition, in some of the subjects, the motility, measured as swimming velocity and distance, was even greater than in the control condition, suggesting a health-promoting effect of the peptides. These last results point out the efficacy of the isolated PF also in vivo and further validate the positive action of the bioactive peptides on the overall animal wellness, increasing the translatability potential of our findings. In fact, tackling oxidative stress could be beneficial to contrast the aging process, which is usually associated with an increased oxidative state, but it can also be an adjuvant approach in non-communicable diseases, which affects a large part of population especially in Western countries [8]. In addition, the antioxidant activity of the MP fraction could also be beneficial as a functional ingredient for food preservation, potentially extending the shelf-life of the products. After these results, as a future perspective of this work, it would be of relevance to explore the long-term health effects of bioactive peptide consumption, investigate the use of MP in functional food formulations, and study the interactions between these peptides and gut microbiota.

## 5. Conclusions

The agri-food industry produces great amounts of by-products from food processing, which are often difficult to dispose of and have a high environmental impact. Regarding the dairy farm sector, the production of milk derivatives is associated with the creation of food by-products as MP. The latter is poorly utilized for animal feeding. In recent years, in the context of sustainability and circular economy, new methods for its valorization have been researched especially to extract lactose and minerals. So far, the peptide component of MP instead has been poorly explored. In this frame, with this research, we successfully isolated and identified new peptides with bioactive properties from MP. The peptide fractions, and particularly the 5–30% ACN fraction, were endowed with antioxidant properties in scavenging tests. Furthermore, the PF was able to protect cultured intestinal cells from oxidative stress. The mechanism of antioxidant action has been explored and correlated to the activation of the Nrf2/Keap1 pathway. In fact, the translocation of the transcription factor to the nucleus and the increased expression of downstream antioxidant enzymes were observed. The bioactive fraction was also able to increase mitochondrial oxygen consumption, promoting aerobic metabolism in the same cellular model. Noteworthy, the PF was effective in contrasting cold stress, improving the animals’ swimming capacity in the zebrafish model. This result indicates that the newly isolated bioactive peptides are effective also when administered to a whole organism. This latter observation allows for future research on the potential application of these peptides for the development of functional foods. Indeed, with the aging of the population associated with a better knowledge of the importance of correct nutrition, the request of health-promoting foods is increasing. Thus, the current research opens to commercial potential of the bioactive peptides for use in dietary supplements and functional foods. In conclusion, this research opens to the valorization of milk permeate, a dairy-farm by-product, via the recovery of health-promoting bioactive peptides endowed with antioxidant activity that was observed both in vitro and in vivo.

## Figures and Tables

**Figure 1 antioxidants-13-01221-f001:**
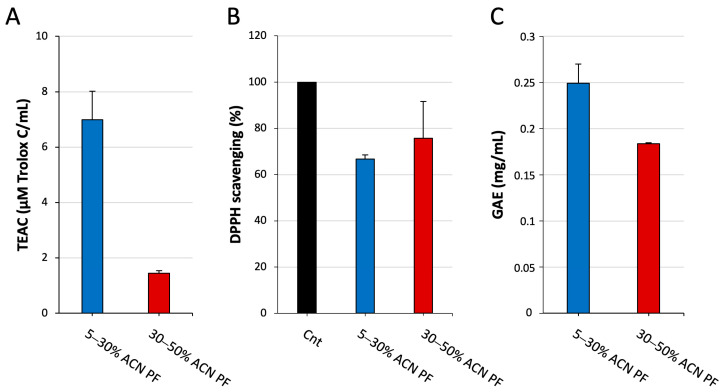
Antioxidant activity of PFs obtained from MP. (**A**) ABTS scavenging activity is reported as Trolox C equivalent antioxidant capacity (TEAC); (**B**) DPPH scavenging assay; results are reported as percentage with respect to the control (Cnt); (**C**) estimation of total phenolic content expressed as gallic acid equivalent (GAE). Results are the mean ± SD of three replicates.

**Figure 2 antioxidants-13-01221-f002:**
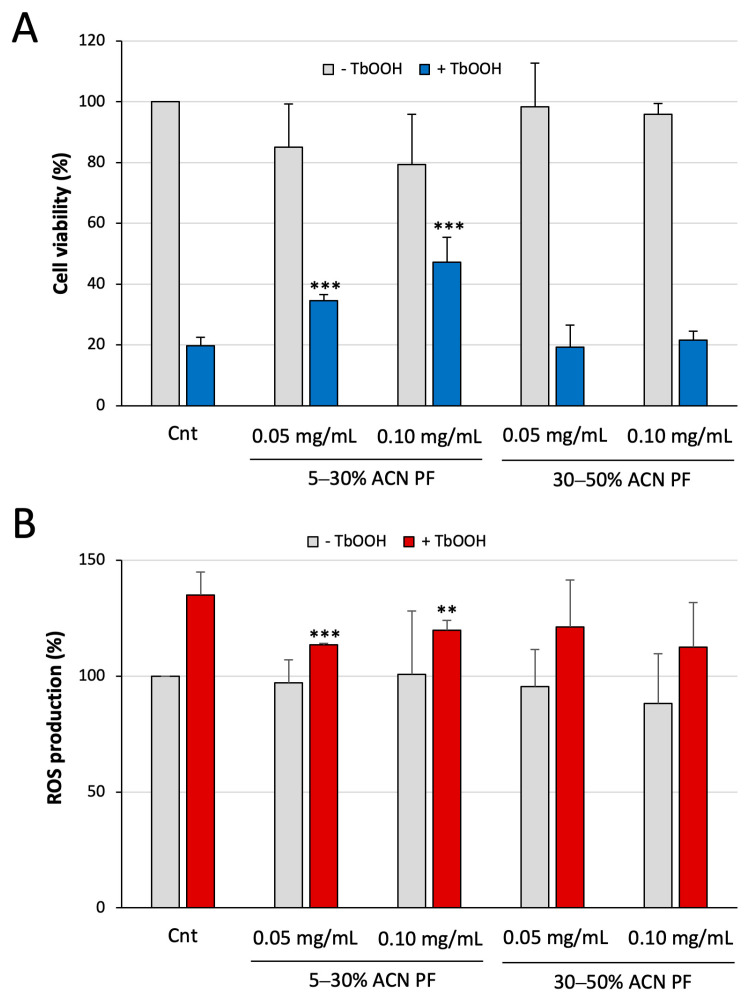
Protective effect of PFs obtained from milk permeate on oxidative stress induction in Caco-2 cells. (**A**) Effects of the PFs on cell viability. Caco-2 cells were treated with the indicated fractions for 24 h and oxidative stress was induced by 180 µM TbOOH (for 18 h). Results are shown as percentage of cell viability with respect to the Cnt; (**B**) Estimation of ROS production in Caco-2 cells treated with the indicated PFs for 24 h in the absence (grey) or presence (red) of 300 µM TbOOH and expressed as percentage with respect to the Cnt. Results are the mean ± SD of three replicates ** *p* < 0.01; *** *p* < 0.001.

**Figure 3 antioxidants-13-01221-f003:**
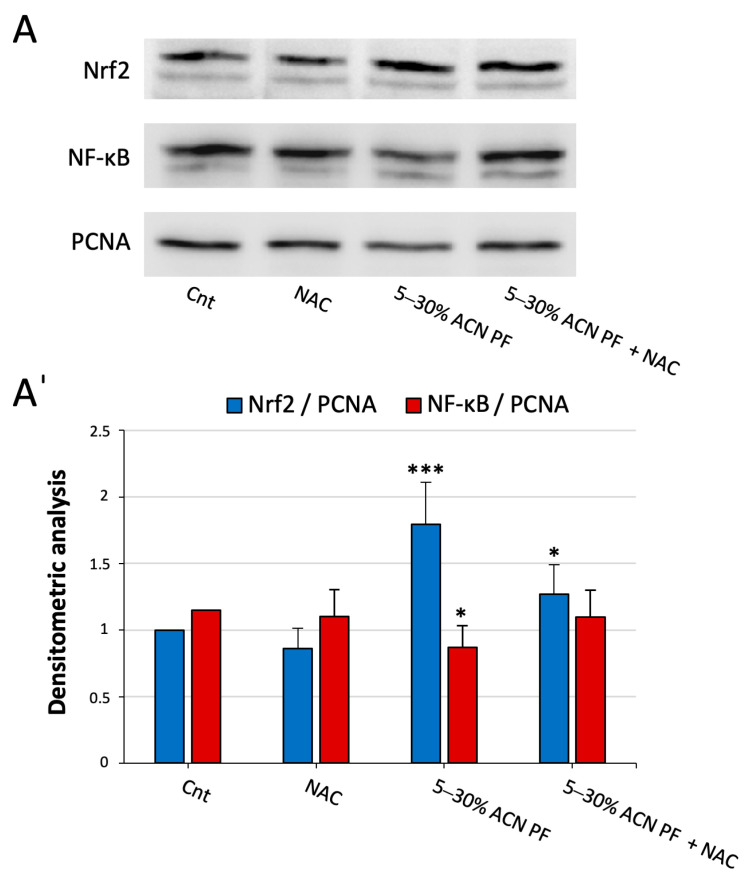
Nrf2 and NF-κB levels in the nuclear fraction of Caco-2 cells treated with 5–30% ACN PF from MP. Nuclear fractions of cells treated with 0.05 mg/mL of 5–30% ACN PF and/or with 2 mM NAC for 24 h were extracted, and Western blot analysis was carried out to estimate Nrf2, NF-κB, and PCNA levels. (**A**) Representative WB of protein expression in Caco-2 cells in the different conditions. (**Aˈ**) Quantitative analysis of the WB after normalization using PCNA as a nuclear loading control. Results are the mean ± SD of three replicates * *p* < 0.05; *** *p* < 0.001.

**Figure 4 antioxidants-13-01221-f004:**
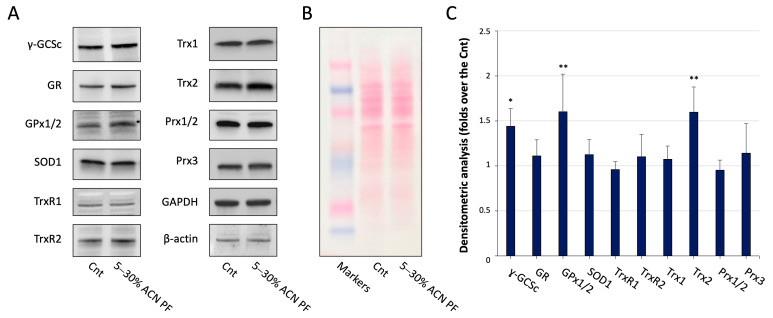
Expression of antioxidant enzymes in Caco-2 cells treated with 5–30% ACN PF from MP. Cells were treated with 0.05 mg/mL of 5–30% ACN PF for 24 h. Afterwards, cells were lysed, and WB analysis was carried out. (**A**) Representative images of protein expression of the various enzymes in Caco-2 cells via WB; (**B**) Ponceau S staining reporting the protein loading; (**C**) Quantitative analysis of the Western blot after normalization using GAPDH and β-actin as loading controls. Results are the mean ± SD of three replicates * *p* < 0.05; ** *p* < 0.01.

**Figure 5 antioxidants-13-01221-f005:**
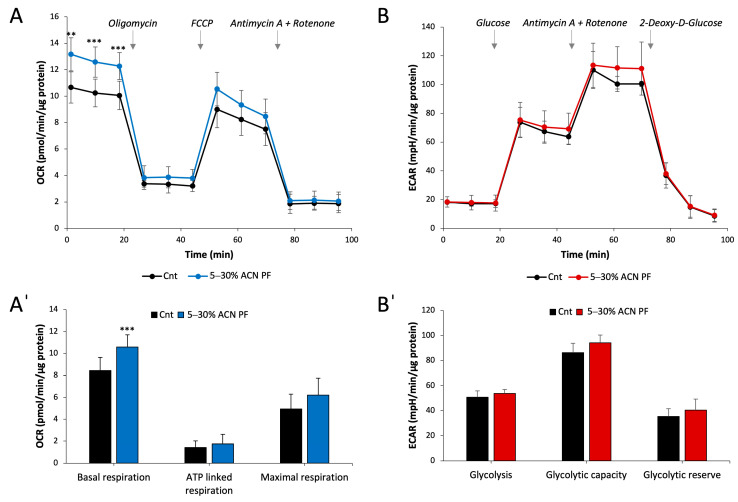
Oxygen consumption rates and glycolytic activity of Caco-2 cells treated with the PF. Caco-2 cells were treated with 0.05 mg/mL of the 5–30% ACN PF for 24 h. (**A**) The oxygen consumption rates (OCRs) were assessed using the Seahorse Xfe24 analyzer as described in the Section 2. Basal respiration and respiratory capacity in the presence of sequential addition of 1 µM oligomycin, 0.5 µM FCCP and the combination of 1 μM antimycin A + 1 µM rotenone, was measured. (**Aˈ**) Basal, ATP-linked and maximal respirations are shown as the mean ± SD of 5 experiments, ** *p* < 0.01, *** *p* < 0.001. (**B**) Cellular glycolytic activity was determined in the presence of sequential addition of 10 mM glucose, 1 μM antimycin A + 1 µM rotenone, and 2-deoxy-D-glucose. (**Bˈ**) Glycolysis, maximal glycolytic capacity, and glycolytic reserve are reported as the mean ± SD of 3 experiments.

**Figure 6 antioxidants-13-01221-f006:**
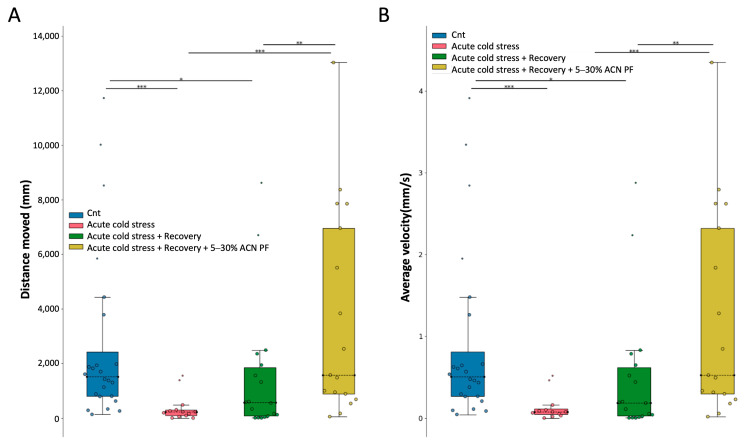
Effects of the 5–30% ACN PF in vivo on zebrafish larvae under cold stress conditions. Zebrafish larvae were divided into four groups: control (placed in 28 °C water) (blue box), acute cold stressed (placed in 10 °C water for 5 min) (pink box), acute cold stressed + 30 min recovery (green box), and acute cold stressed + 30 min recovery in the presence of the PF (yellow box). The motility of the four groups of larvae was assessed using the DanioVision system. (**A**) Total swimming distance; (**B**) Average swimming speed. (*n* = 75, distributed as 25 controls, 13 cold stress, 19 cold stress + recovery, and 18 cold stress + recovery + 5–30% ACN PF), * *p* < 0.05, ** *p* < 0.01, *** *p* < 0.001.

**Table 1 antioxidants-13-01221-t001:** Composition of milk permeate.

Parameter	Value
**Total proteins**	0.37 g/100 g
**Total fats**	0 g/100 g
**Lactose**	4.51 g/100 g
**Dry leftover**	4.91% *w*/*w*
**pH**	6.78
**Thiamine**	0.192 ± 0.074 µg/kg

These parameters were analyzed by an external source (CHELAB, Resana, TV, Italy).

**Table 2 antioxidants-13-01221-t002:** Antioxidant activity of milk permeate assessed with ABTS and DPPH scavenging assays and via estimation of total phenolic content.

Assay	Results
ABTS scavenging	236.25 ± 13.66 TEAC (µM Trolox C/mL)
DPPH scavenging	37.72 ± 7.65 (%/100 µL)
Total phenols	53.84 ± 3.80 GAE (mg/100 mL)

## Data Availability

The original contributions presented in the study are included in the article/Appendix A, further inquiries can be directed to the corresponding authors.

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
