# Peer review of "By-Products Valorization: Peptide Fractions from Milk Permeate Exert Antioxidant Activity in Cellular and In Vivo Models"

_antioxidants, 2024, doi:10.3390/antiox13101221_

Round 1

Reviewer 1 Report

The manuscript covers an interesting and relevant topic - the use of permiate from ultrafiltration in the dairy industry for the production of a higher biological value product. A complex experiment was carried out: the composition and antioxidant properties of milk permiate were determined, chromatographic purification of peptides from milk permeate were done and two fractions was obtained, the peptides present in the different fractions were identified, antioxidant properties of peptide fractions in vitro and in vivo were studied. Also impact of the peptide fraction on the cellular metabolism and zebrafish larvae were investigated. The results of the study may be useful in the future for the production of higher value-added products from by-products of dairy industry.

But there are some comments for the authors:

The results section repeats information already provided in the Materials and Methods section (For example 324 line 220 nm, 328 line LTQ-Orbitrap XL mass spectrometer). 

Figure 3 – title and legend should be under figure, not on the top.

Citations and reference lists are not in accordance with the Antioxidants journal requirements.

Reviewer 2 Report

The manuscript entitled “By-products valorization: peptide fractions from milk permeate exert antioxidant activity in cellular and in vivo models” aims to reduce the environmental impact of the by-product from milk permeate 2, searching for bioactive components endowed with antioxidante and anti-inflammatory properties. The manuscript is well presented and contains important information, which merits publication. Moreover, important concluding remarks were made. The presentation, organization and length of the paper are satisfactory. In general, the manuscript has a good technical quality and clarity of presentation. For all these reasons, I suggest that the manuscript is acceptable for publication after some corrections.

1. I recommend that the authors incorporate experimental data in Abstract section.

2. How is this system different to other reports to merit publication? Please, report in Introduction section.

3. Tables 3 and 4 should be incorporated to a Supplementary Material (excessive information/data).

4. A comparative study with previous reports is highly required.

5. I recommend a strong improvement in resolution/ of the list of abbreviations (bad quality).

6. References should be standardized according to Instruction for authors.

The manuscript entitled “By-products valorization: peptide fractions from milk permeate exert antioxidant activity in cellular and in vivo models” aims to reduce the environmental impact of the by-product from milk permeate 2, searching for bioactive components endowed with antioxidante and anti-inflammatory properties. The manuscript is well presented and contains important information, which merits publication. Moreover, important concluding remarks were made. The presentation, organization and length of the paper are satisfactory. In general, the manuscript has a good technical quality and clarity of presentation. For all these reasons, I suggest that the manuscript is acceptable for publication after some corrections.

1. I recommend that the authors incorporate experimental data in Abstract section.

2. How is this system different to other reports to merit publication? Please, report in Introduction section.

3. Tables 3 and 4 should be incorporated to a Supplementary Material (excessive information/data).

4. A comparative study with previous reports is highly required.

5. I recommend a strong improvement in resolution/ of the list of abbreviations (bad quality).

6. References should be standardized according to Instruction for authors.

Reviewer 3 Report

The primary goal of the research should be more explicitly stated in the abstract and introduction to ensure that readers can easily follow the focus of the study.

Clearly articulate how the recovery strategies introduced in this study differ from existing methods, emphasizing the innovation and potential impact on the field.

Expand on the implications of valorizing milk permeate for sustainability in the dairy industry. Discuss how the research contributes to waste reduction and the potential for developing functional foods that promote health.

Abstract

Ensure the primary goal of the research is clearly stated and aligned with the other objectives, making the focus of the study easy to follow.

Clearly articulate how the "novel recovery strategies" introduced in the study differ from existing methods, highlighting the innovation.

Use confident phrasing, like "the results demonstrated" rather than "the results pointed out," to present the findings with greater authority.

Provide more specific details about the outcomes of the zebrafish model, particularly the significance of protecting the organism from oxidative stress.

Introduction

Make the environmental and economic impact of milk permeate disposal more explicit and connect it clearly to the need for valorization.

Clearly highlight the gap in current research or practices that this study addresses, emphasizing how milk permeate valorization fills this gap.

Emphasize what is novel about the peptides derived from milk permeate, differentiating them from other sources and advancing current knowledge.

Provide a concise explanation of the Keap1/Nrf2 and NF-κB pathways, focusing on their relevance to the study's findings without overwhelming the reader.

End with a clear statement about the broader impact of the study on food industry sustainability and potential health applications.

Materials and Methods

Clearly mention the purity of the chemicals and reagents (e.g., ≥99%) and provide catalog numbers or batch numbers for reproducibility.

Specify the number of replicates (e.g., triplicate or duplicate) performed for each assay to assess reproducibility and statistical reliability.

Add more detailed specifications for the equipment used, such as model numbers for spectrophotometers, plate readers, and other key instruments. This enhances reproducibility.

Include more details on the nuclear and cytosolic fractionation process, such as the specific buffers used and the final concentration of each component.

Describe how calibration curves were constructed for the ABTS and DPPH assays, including any specific concentrations used and the number of replicates to validate results.

Results

To enhance the visual clarity of Table 1, please consider removing some horizontal lines. This adjustment will help reduce clutter and make the table more aesthetically pleasing and easier to read.

For all figures, please indicate statistical differences by labeling the bars with the same letters for groups that are not significantly different and different letters for groups that show a significant difference. This approach will enhance clarity and provide a quick visual reference for understanding the significance of the data.

For all the figures, the figure labels should be placed under the figures for consistency. However, for Figure 3, the label is currently positioned above the figure. Please move the label for Figure 3 to be below the figure to maintain uniformity throughout the document.

Ensure that all terms and methods used for the analysis are clearly defined. For instance, briefly explain the significance of the Dumas combustion method and why it was chosen for protein evaluation.

Include statistical data (e.g., standard deviations, confidence intervals) for the parameters reported in Table 1. This adds credibility to the results and allows for better interpretation.

When mentioning that some vitamins were lower than the limit of detection, specify what this limit is. This will give readers a clearer understanding of the sensitivity of your analysis.

After presenting the composition data, discuss the implications of these findings regarding the nutritional or functional properties of milk permeate. This connects the data to the broader research context.

Consider adding a figure or diagram that summarizes the composition of milk permeate to enhance visual comprehension, especially for complex data.

Emphasize the most significant findings regarding antioxidant activity and phenolic content early in the section to capture the reader's interest.

Ensure smooth transitions between sections, particularly from the composition of milk permeate to the in vitro estimation of antioxidant activity, to enhance readability.

Include a brief discussion on the biological relevance of the antioxidant activities observed, emphasizing how they could potentially impact health or food preservation.

Elaborate on the mechanisms by which the 5–30% ACN PF protects Caco-2 cells, including potential pathways involved in ROS reduction.

Present a dose-response curve for the protective effects of the 5–30% ACN PF in Caco-2 cells to provide more comprehensive insights into its efficacy.

Compare the effects of the PF treatment with other known antioxidants to highlight the unique action of the PF in regulating the Keap1/Nrf2 pathway.

Ensure all figures are clearly labeled and include legends that explain the significance of the findings, making it easier for readers to interpret the data.

Discussion

Expand on the implications of valorizing milk permeate (MP) in the context of a circular economy. Discuss how the extraction of bioactive compounds not only reduces food waste but also contributes to sustainability and environmental health.

Discuss the implications of the observed effects on cellular oxygen consumption and glycolytic rates in a broader metabolic context. Explore how these findings could influence dietary recommendations or therapeutic strategies for metabolic disorders.

Strengthen the discussion on the significance of the in vivo results using zebrafish. Highlight how the findings could inform further research into the potential health benefits of bioactive peptides in various animal models or human studies.

Suggest specific future research directions that could build on the current findings. This could include exploring the long-term health effects of bioactive peptide consumption, investigating the use of MP in functional food formulations, or studying the interactions between these peptides and gut microbiota.

Conclude with a discussion on the broader implications of this research for food science and nutrition, particularly how it may influence the development of functional foods aimed at promoting health and well-being.

Conclusion

Emphasize the role of valorizing milk permeate in addressing waste management and reducing the environmental impact of the dairy industry, highlighting its importance in the context of sustainability and the circular economy.

Clearly outline specific future research directions, such as exploring the long-term health effects of the isolated peptides in human trials and their potential applications in functional foods.

Discuss the commercial potential of the bioactive peptides for use in dietary supplements and functional foods, emphasizing market demand for natural health-promoting ingredients.
